# Mint essential oil: A natural and effective agent for controlling house dust mites

Haiming Cai[1], Xu Zhang[1], Zhibin Lin[1], Shanshan Li[2], Huiquan Lin[1]*, Yongwen Lin[3]*

**1** Zhangzhou Affiliated Hospital of Fujian medical University, Zhangzhou, China, **2** Institute of Subtropical Agriculture, Fujian Academy of Agriculture Sciences, Zhangzhou, China, **3** Zhangzhou Institute of Technology, Zhangzhou, China

\* linyongwen@fjzzit.edu.cn (LY); linhuiquan@126.com (LH)

## Abstract

Conventional methods of house dust mite control often involve chemical pesticides, raising concerns about their potential hazards. Mint essential oil presents a natural and eco-friendly alternative for managing house dust mite infestations. In this study, ten varieties of mint plants were cultivated, and their essential oils were extracted through steam distillation. The toxicity of these mint essential oils and their main compounds on adult house dust mites was assessed using contact+fumigant mortality bioassays and vapour-phase mortality bioassays. A repellent bioassay was also conducted to evaluate the repellent effects of mint oils and main compound on house dust mites. The toxicity of mint essential oils varied among the different varieties, with some demonstrating higher potency in eradicating house dust mites. Mint oils showed both acaricidal action and repellent effects on house dust mites, with certain varieties exhibiting stronger efficacy. Linalool as active compounds was identified as key contributors to the acaricidal properties of mint essential oil. Mint essential oil, particularly certain varieties rich in active compounds, shows promise as a natural and effective agent for controlling house dust mites. Its dual functionality in killing and repelling house dust mites, along with its environmentally friendly nature, make it a viable alternative to chemical pesticides for house dust mite management. Further research on the specific mechanisms of action and long-term effectiveness of mint essential oil in house dust mite control is warranted to explore its full potential as a sustainable pest management solution.

## 1. Introduction

Mint plant, *Mentha* L., is a perennial herb with a refreshing aroma and flavor [1–3]. It is widely cultivated for its culinary, medicinal, and ornamental uses [1,4–6]. Mint plants have square stems, serrated leaves, and small flowers in shades of white, pink, or purple. Common varieties of mint include peppermint, spearmint, and chocolate mint, each with its own distinct flavor profile [7]. Mint essential oil is derived from the steam distillation of the leaves of the mint plant, particularly peppermint (*Mentha X piperita*) or spearmint (*Mentha spicata*) [8]. Mint essential oil has gained significant attention for its strong, refreshing aroma and cooling properties [8]. Mint essential oil is also commonly used in controlling insect pest.

**Data availability statement:** The data underlying the results presented in the study

are available from https://doi.org/10.6084/m9.figshare.25865035.

**Funding:** This study was supported by the Doctoral fund of Zhangzhou Institute of Technology (grant number ZZYB2207), and the Special Project of Public Welfare Scientific Research Institutes of Fujian Provincial Science and Technology Department (2023R1028001).

**Competing interests:** The authors have declared that no competing interests exist.

Mint is known for its insect-repelling properties, including its ability to deter and kill insect pest and mites [9,10]. Mint leaves or mint essential oil can be used to create a natural mite repellent spray. Mint essential oil showed the highest repellent activity against yellow mealworm (*Tenebrio molitor*) Larvae when compared to other plant oils [11]. The strong aroma of mint acts as a deterrent to mites, disrupting their sensory receptors and driving them away [12]. Additionally, the compounds found in mint, such as menthol and limonene, have insecticidal properties that can help kill mites upon contact [13,14]. Presented research showed that the essential oils of *Mentha piperita* (Lamiaceae) and its main compounds--menthol can be considered potential acaricides for *Tetranychus urticae* Kogan and *Eutetranychus orientalis* (Klein) [15,16]. This natural approach is often preferred by those seeking chemical-free solutions for mite control. However, the recently researches presented the toxicity of mint essential oils on the field but not inside house.

House dust mites (*Dermatophagoides pteronyssinus*) are minuscule arachnids that flourish in indoor settings, predominantly inhabiting bedding, upholstery, and carpets [17–19]. These tiny creatures are notorious for triggering allergies and asthma symptoms, especially in individuals with sensitivities [20,21]. Conventional approaches to dust mite management typically rely on chemical pesticides, which, despite their effectiveness, raise concerns due to potential health hazards associated with their usage [22–24]. As a result, there is a growing interest in exploring alternative and safer methods to combat dust mite infestations while ensuring the well-being of occupants and the environment [25–27].

This study aims to evaluate the effectiveness of mint essential oils in controlling house dust mites. The toxicity of mint essential oils and its main compounds on house dust mites in controlled environments were assessed, and we provided valuable insights into its potential use as a natural and eco-friendly solution for managing house dust mite infestations.

## 2. Materials and methods

### 2.1. Mite

*Dermatophagoides pteronyssinus* were collected from a bedroom in Zhangzhou city and reared for 6 months without contact with any recognized acaricides. The mites were cultured in petri dishes (8.5 cm in diameter, 5.5 cm in depth) filled with a sterilized diet (a mixture of wheat bran and dried yeast in a 1:1 ratio by weight) at 25 ± 1 °C and 75% relative humidity in darkness. The dried yeast used in the diet was sourced from Angel Yeast Co., Ltd in Yichang, China.

### 2.2. Cultivation of mint plant

Ten varieties of mint plants (refer to Table 1) were cultivated in a greenhouse in Zhangzhou, China for 10 months prior to their utilization in the experiment. After that, the aerial parts of all mint samples were harvested during the blooming stage and promptly cut into 15 cm segments for essential oil extraction.

### 2.3. Extraction of essential oils

The essential oils from the aerial parts of 10 distinct mint varieties were individually extracted through steam distillation according to previous study [28]. Each one kg mint sample was subjected to steam distillation for 90 minutes. Post-extraction, any water and impurities in the essential oil were eliminated with anhydrous sodium sulfate (Xilong Scientific LLC., Guangzhou), and the oil was stored at 4 °C for subsequent analysis.

**Table 1. Ten varieties of mint used in this study and their respective abbreviations.**

| Varieties | Abbreviation |
|---|---|
| *Mentha canadensis* L. | MC1 |
| *M. aquatica* Citr. | MC2 |
| *M. suaveolens* Ehrhart | MC3 |
| *M. piperita* 'Chocolate' | MC4 |
| *M. piperita* 'Grapefruit' | MC5 |
| *M. japonica* (Miq.) Makino | MC6 |
| *M. crispata* Schrader ex Willd. | MC7 |
| *M. piperita* 'Candy Mint' | MC8 |
| *M. arvensis* 'Banana' | MC9 |
| *M. aquatica* L. | MC10 |

## 2.4. Chemicals

There were 3 chemicals used in this study. Of these, analytically pure of limonene and menthol were purchased from Sigma-Aldrich LLC., and analytically pure of ethyl alcohol which used to dilute essential oil, limonene and menthol were purchased from Xilong Scientific LLC (Guangzhou, China).

## 2.5. GC-MS

For Gas Chromatography-Mass Spectrometry (GC–MS) analysis, the samples were dissolved in chloroform at a ratio of 50 μL oil:1 mL chloroform and injected into the GC system (Agilent Technologies). The setup included a gas chromatograph (7890B) with a mass spectrometer detector (5977A) at NRC, utilizing an HP-5MS column (30 m × 0.25 mm internal diameter and 0.25 μm film thickness). Helium served as the carrier gas at a flow rate of 3.0 mL/min with a split ratio of 1:10 and an injection volume of 1 μL. The temperature programme comprised the following steps: initial 40 °C for 1 min, ramping at 10 °C per minute to 200 °C for 1 min, further ramping at 20 °C per minute to 220 °C for 1 min, and a final ramp at 30 °C per minute to 320 °C for 3 min. The injector and detector temperatures were maintained at 250 °C and 320 °C, respectively. Mass spectra were obtained using electron ionization (EI) at 70 eV with a spectral range of m/z 30–550 and a solvent delay of 2.5 min. The quad operated 150 °C above the mass temperature of 230 °C. Identification of the 10 kinds of essential oils constituents was achieved by comparing the spectrum fragmentation pattern with data from the Wiley and NIST (2017) Mass Spectral Library.

## 2.6. Acaricidal bioassay

An established fabric-circle contact+fumigant mortality bioassay, as detailed in prior research [26], was employed to assess the toxicity of mint oils on adult house dust mites. Various concentrations of each test substance in 50 μL of ethanol were administered to 4.5 cm diameter black cotton-fabric circles, resulting in mint oil quantities of 300, 150, 75, 37.5, 18.75, 9.325, and 5 μL/cm² applied on the fabric. After air-drying for 1 minute, these fabric circles were placed on the lower section of a 4.5 cm × 1 cm petri dish. Groups of 25–30 adult mites (both sexes, 5–8 days old) were individually introduced onto the treated fabric circles. The petri dishes were sealed with Bemis Parafilm M (Neenah, WI). Benzyl benzoate (Xilong Scientific LLC., Guangzhou) was used as a positive control (since it is known as a recommended acaricide), while negative controls consisted of 50 μL of ethanol only. Treated and control mites

were maintained under standard colony conditions, and mortality rates were assessed 24 hours post-treatment under a dissecting microscope (×20). Mites were classified as deceased if no movement was observed in their body or appendages when gently prodded with a fine wooden dowel. Each treatment was replicated three times using 30 adults per replication.

## 2.7. Bioassay of vapour-phase mortality

In this section, closed and open container methods were employed to assess whether the lethal effects of mint oils on adult house dust mites were due to contact or fumigant actions [26]. Thirty adult mites (both sexes, 5–8 days old) were individually placed on untreated cotton-fabric circles in petri dishes sealed with lids featuring a fine wire screen over a central hole. Filter papers treated with approximately three times the contact+fumigant $LC_{100}$ values of each compound were positioned on top of the wire screen to prevent direct mite contact. The petri dishes were sealed with either a solid lid (closed container method) or a lid with a central hole (open container method, $\Phi 0.5\,cm$) to assess potential vapor-phase toxicity. Benzyl benzoate was used as a positive control, while negative controls consisted of 50 μL of ethanol only. The petri dishes were kept in a closed container (0.5 m³) and covered with black cloth. Mortality rates were assessed 24 hours post-treatment, as outlined above. Each bioassay was replicated three times.

## 2.8. Bioassay of repellent

In a modified version of a previous study [15], a repellent bioassay was conducted. Eleven concentrations of main compounds and sublethal concentrations ($LC_{15}$) of 10 types of mint oils were used. A black cloth piece (4.5 cm diameter, 100% cotton) treated with 100 μL of compounds in a 22.5 mm × 22.5 mm quadrant using a micropipette was affixed in a petri dish (9.5 cm diameter) with double-sided tape. After placing the growth medium on filter paper for 5 minutes, 30 adult mites (without food) were placed on the filter paper initially placed on the media and were then transferred to the compound-treated section using a brush to avoid damage, and the petri dishes were thereafter closed immediately to avoid escape. A control group was treated with 100% ethanol. Treated and control mites were kept at 25 ± 1 °C and 75% relative humidity in darkness. Mite avoidance, indicated by repellency from the chemical treatment area, was assessed after 2 hours as per previous protocols. Each treatment was replicated three times.

## 2.9. Statistical analysis

Control mortality was adjusted using Abbott's formula as follow:
adjusted mortality = tested mortality − control mortality.
Concentration-mortality data underwent probit analysis to determine the $LC_{50}$ values for each group. Significance between the $LC_{50}$ values of different species and treatments was established when the 95% confidence limits (CLs) did not overlap. Mortality percentages were transformed into arcsine square root values for analysis of variance. The Bonferroni multiple-comparison method was employed to identify significant differences among treatments, while a t-test was used to assess variations between two treatment methods. Results are presented as means ± standard deviation (SD) of untransformed data. All the statistical analyses were performed using GraphPad Prism 9 software.

## 3. Results and discussion

### 3.1. GC-MS detection for the essential oils of 10 varieties of mints

GC-MS analysis of the 10 varieties of mint essential oils revealed the presence of four key compounds (Table 2). L-menthone was identified as the predominant compound in MC1, MC3, MC6, and MC7, while D-limonene was predominant in MC2, MC4, MC8, and MC9. Linalool was the primary compound in MC5 and MC10.

### 3.2. Toxicity of mint oils of 10 var. to house dust mites

The toxicities of 10 kinds of oils from mint plant to adult house dust mites are listed in Table 3. As judged by 24 h $LC_{50}$ values, these oils can be categorized into three levels. MC5 and MC10 oils (74.65 and 34.72 µl cm⁻² respectively) were in the first level, approximately two times less toxic than benzyl benzoate (34.72 µl cm⁻²). MC8, MC9, MC7, MC6, and MC1 oils (140.88, 178.76, 179.15, 195.31, and 196.93 µl cm⁻² respectively) were in the second level,

**Table 2. Most abundant compound identified from each of the ten varieties of mint following steam distillation and GC-MS analyses.**

| Varieties | Main compound | Retention times (min)[a] | Percentage area% |
|---|---|---|---|
| MC1 | L-menthone | 17.361 + 22.435 | 62.98 ± 4.35 |
| MC2 | D-limonene | 10.278 | 45.01 ± 3.55 |
| MC3 | L-menthone | 17.361 + 22.435 | 55.16 ± 4.46 |
| MC4 | D-limonene | 10.33 | 50.54 ± 0.86 |
| MC5 | linalool | 14.295 | 65.92 ± 2.05 |
| MC6 | L-menthone | 17.361 + 22.435 | 50.11 ± 0.43 |
| MC7 | L-menthone | 17.361 + 22.435 | 49.17 ± 0.29 |
| MC8 | D-limonene | 10.295 | 58.32 ± 6.77 |
| MC9 | D-limonene | 10.278 | 40.51 ± 2.6 |
| MC10 | linalool | 14.215 | 48.92 ± 0.88 |

[a]There were two retention times of L-menthone in MC1, MC6 and MC7.

**Table 3. Toxicity of essential oils of 10 varieties mint against adult house dust mites using a contact+fumigant mortality bioassay with 24 h exposure.**

| Varieties[a] | $LC_{50}$(µl/cm²) (95%CL) | $LC_{15}$(µl/cm²) | $LC_{100}$(µl/cm²) | Slop ± SD | $\chi^{2}$[b] | P value[c] |
|---|---|---|---|---|---|---|
| MC1 | 196.93 (140.57–328.73) | 59.08 | 393.86 | 0.2539 ± 0.03666 | 12.69 | 0.9977 |
| MC2 | 259.88 (218.05–321.34) | 77.96 | 519.75 | 0.1924 ± 0.01328 | 4.597 | 0.9999 |
| MC3 | 275.48 (202.68–429.92) | 82.64 | 550.96 | 0.1815 ± 0.02347 | 8.128 | 0.9985 |
| MC4 | 215.8 (154.04–360.23) | 64.74 | 431.59 | 0.2317 ± 0.03345 | 11.58 | 0.9977 |
| MC5 | 74.65 (50.55–142.65) | 22.39 | 149.3 | 0.6698 ± 0.115 | 19.91 | 0.9957 |
| MC6 | 195.31 (135.06–352.36) | 58.59 | 390.63 | 0.256 ± 0.04112 | 14.24 | 0.9966 |
| MC7 | 179.15 (116.82–383.73) | 53.74 | 358.29 | 0.2791 ± 0.0536 | 18.56 | 0.9935 |
| MC8 | 140.88 (98.64–246.43) | 42.27 | 281.77 | 0.3549 ± 0.05474 | 18.95 | 0.9971 |
| MC9 | 178.76 (114.47–407.83) | 53.63 | 357.53 | 0.2797 ± 0.05659 | 19.59 | 0.9922 |
| MC10 | 65.15 (41.74–148.41) | 19.54 | 130.29 | 0.7675 ± 0.1551 | 26.85 | 0.9922 |
| Benzyl benzoate | 34.72 (26.14–51.65) | 10.42 | 69.44 | 1.44 ± 0.1701 | 14.72 | 0.9989 |

[a]MC1-MC10 represented *M. canadensis* L., *M. aquatica* Citr., *M. suaveolens* Ehrhart, *M. piperita* 'Chocolate', *M. arvensis* 'Banana', *M. japonica* (Miq.) Makino, *M. crispata* Schrader ex Willd., *M. aquatica* L., *M. piperita* 'Grapefruit' and *M. piperita* 'Candy Mint'.

[b]Pearson $\chi^{2}$, goodness-of-fit test.

[c]t-test, possibility of the result.

while MC4, MC2, and MC7 oils (215.8, 259.88, and 275.48 µl cm$^{-2}$ respectively) were in the third level. Mortality in the negative control group was below 2%, specifically at 1.11%. The results indicate that mint essential oils exhibit varying levels of toxicity towards adult house dust mites, with certain varieties showing higher efficacy compared to others. This variation in effectiveness may be attributed to differences in essential oil compositions among mint varieties [29,30]. This variability in effectiveness underscores the importance of understanding the chemical composition and concentrations of active compounds in different mint varieties to optimize their acaricidal potential.

### 3.3. Acaricidal action of mint oils

The acaricidal activity of mint oils on adult house dust mites was assessed through a vapour-phase mortality bioassay in two settings (Table 4). Following 24-hour exposure to 200 µl/cm$^2$ of MC5 oil, MC10 oil, and benzyl benzoate, significant differences (P < 0.0001) in mortality rates were noted between the closed container (100% mortality) and open container (15%, 16.67%, and 50% mortality, respectively) treatments. Similar significant differences (P < 0.0001) in the mites' responses to the other eight mint oils were observed between closed and open container treatments. The vapour-phase mortality bioassay revealed significant differences in mortality rates between closed and open container treatments, indicating the importance of fumigant actions in the acaricidal activity of mint oils. This highlights the potential of mint essential oils to combat house dust mites through both contact and fumigant mechanisms, enhancing their efficacy in controlling infestations.

### 3.4. Repellent effect of main compounds of the mint varieties

Figs 1–4 display the repellent effects of mint oils and their main compounds (linalool, limonene and menthol) on house dust mites 2 hours post-treatment, with ethanol serving as the diluent. The absence of dead mites following ethanol treatment indicates that the repellent effect is attributed to the mint oils and main compounds. Linalool at a concentration of 3% resulted in a maximum of 22.33 mites were repelled (occupying 74.44%) (Fig 1),

**Table 4. Fumigant toxicity of essential oils of 10 var. mint against adult house dust mites using a vapour-phase mortality bioassay with 24 h exposure.**

| Varieties[a] | Concentration(µl/cm$^2$) | Mortality% ( ± SD) | | P-value[b] |
|---|---|---|---|---|
| | | Closed | Open | |
| MC1 | 591 | 93.33 ± 3.33 | 5 ± 6.67 | <0.0001 |
| MC2 | 780 | 88.89 ± 6.94 | 5.33 ± 3.85 | <0.0001 |
| MC3 | 826 | 100 | 4.33 ± 6.94 | <0.0001 |
| MC4 | 647 | 97.78 ± 3.85 | 12.67 ± 5.09 | <0.0001 |
| MC5 | 200 | 100 | 15 ± 3.33 | <0.0001 |
| MC6 | 586 | 100 | 5.67 ± 7.7 | <0.0001 |
| MC7 | 537 | 87.78 ± 8.39 | 5.67 ± 6.94 | <0.0001 |
| MC8 | 423 | 92.22 ± 1.92 | 11.33 ± 5.09 | <0.0001 |
| MC9 | 536 | 91.11 ± 8.39 | 4.33 ± 3.85 | <0.0001 |
| MC10 | 200 | 100 | 16.67 ± 5.09 | <0.0001 |
| Benzyl benzoate | 200 | 100 ± 0 | 50.00 ± 6.67 | <0.0001 |

[a]MC1-MC10 represented *M. canadensis* L., *M. aquatica* Citr., *M. suaveolens* Ehrhart, *M. piperita* 'Chocolate', *M. arvensis* 'Banana', *M. japonica* (Miq.) Makino, *M. crispata* Schrader ex Willd., M. aquatica L., M. piperita 'Grapefruit' and *M. piperita* 'Candy Mint'.

[b]According to Student's t-test.

while limonene and menthol exhibited the highest repellent effect less than 25% on house dust mites at a all concentration. Mites remained alive when exposed to over 0.4% linalool, limonene and menthol (Figs 1–3), with the number of surviving mites increasing as the chemical concentrations decreased. Compared to sub-lethal concentrations, MC5, MC4, and MC10 oils repelled 67.77%, 64.43% and 57.77% mites, respectively, significantly more than other mint oils(Fig 4). The study highlights the dual functionality of mint essential oils in managing house dust mite infestations. Not only do these oils possess the ability to kill house dust mites, but they also exhibit significant repellent effects. MC4, MC5 and MC10 essential oils with the sublethal concentration showed strong repellent against house dust mite. In addition to the killing effect observed, the strong aroma of the mint may also disrupt the sensory receptors of house dust mites, consequently deterring their presence and reducing infestation risks [31,32]. The repellent properties of mint essential oils, particularly MC5 and MC10 essential oils contained about 50% linalool, offer a natural and eco-friendly alternative to traditional chemical pesticides for house dust mite control. The repellent effects of mint essential oils and their main compounds, such as linalool, further support their utility in repelling house dust mites. The observed repellency, particularly at specific concentrations, underscores the potential of mint essential oils as a natural and non-toxic solution for deterring house dust mites from indoor environments.

## 4. Conclusion

Essential oil formulations containing active constituents show promise as commercial acaricides for integrated mite management [33–37]. These products offer selectivity, minimal impact on non-target organisms, and environmental friendliness. They can complement biological control methods. Essential oils consist of complex hydrocarbons (mainly terpenoids) and oxygenated compounds (such as alcohols, aldehydes, esters, ketones, oxides, and phenols), which

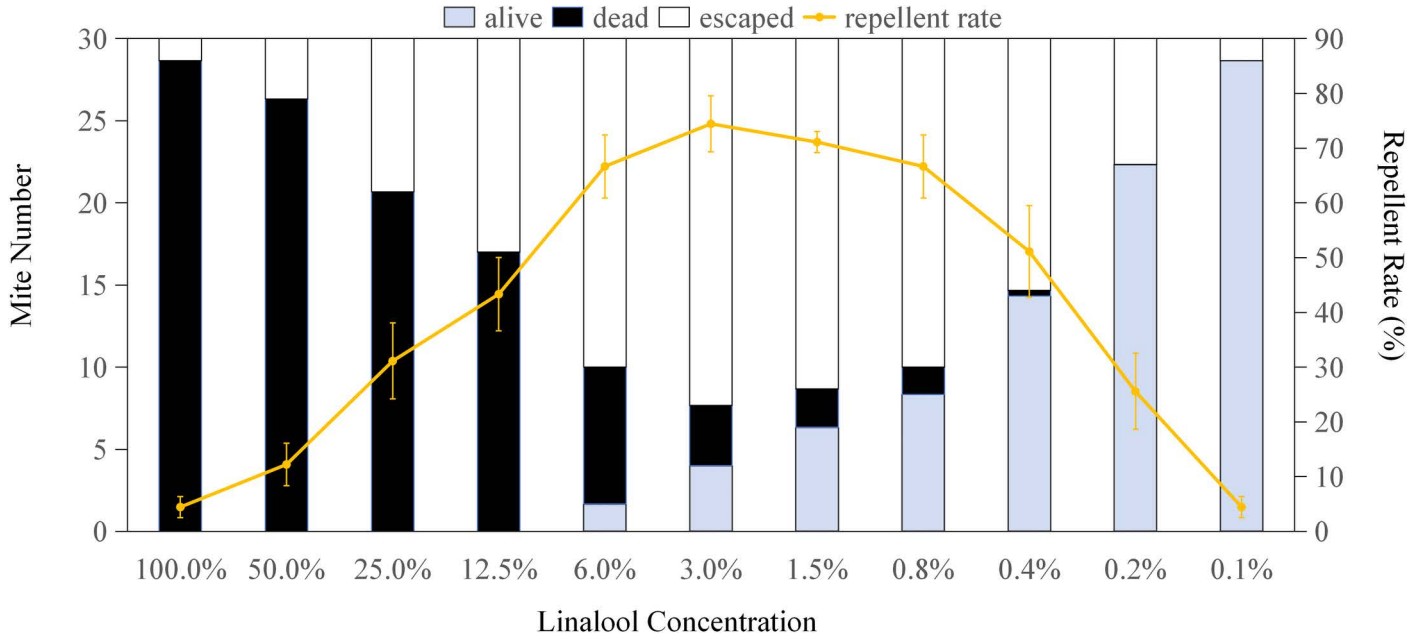

**Fig 1. Repellent effect of linalool on *Dermatophagoides pteronyssinus*.** *Dermatophagoides pteronyssinus* were exposed to linalool (100%, 50%, 25%, 12.5%, 6%, 3%, 1.5%, 0.8%, 0.4%, 0.2% or 0.1%, separately) for 2 hours. The error bars represent standard deviations.

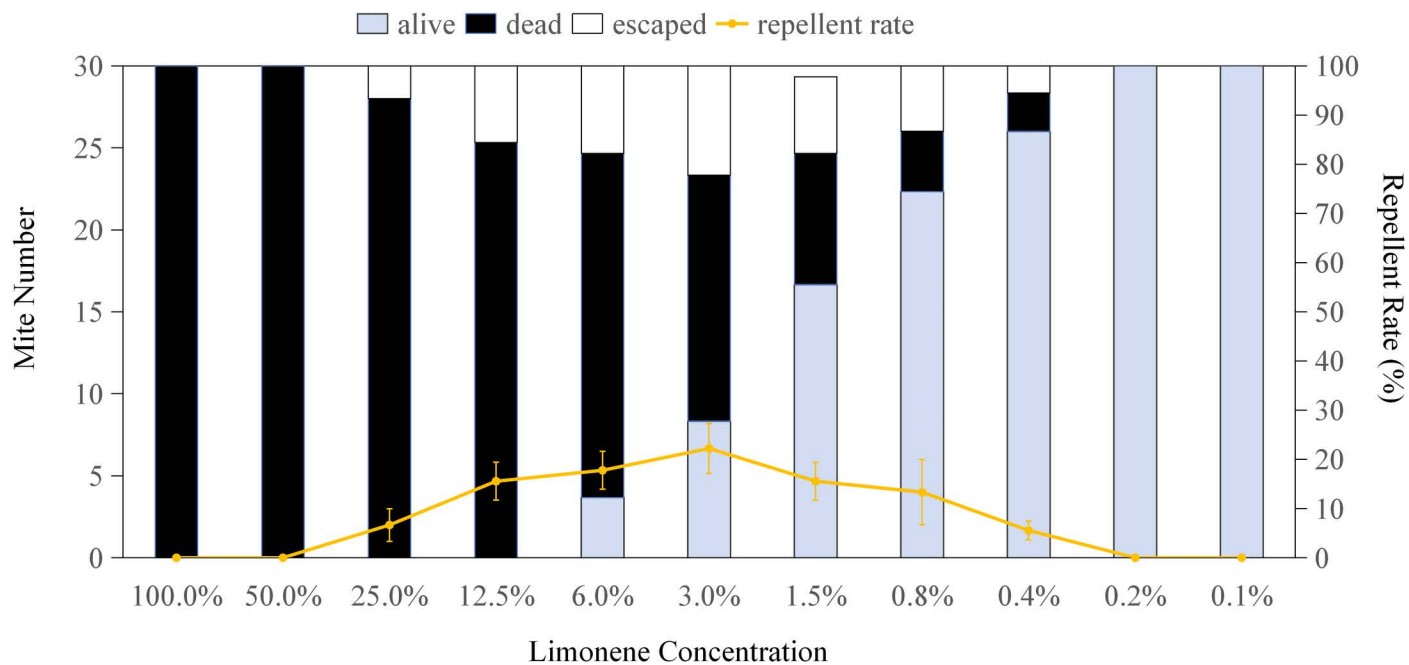

**Fig 2. Repellent effect of limonene on *Dermatophagoides pteronyssinus*.** *Dermatophagoides pteronyssinus* were exposed to limonene (100%, 50%, 25%, 12.5%, 6%, 3%, 1.5%, 0.8%, 0.4%, 0.2% or 0.1%, separately) for 2 hours. The error bars represent standard deviations.

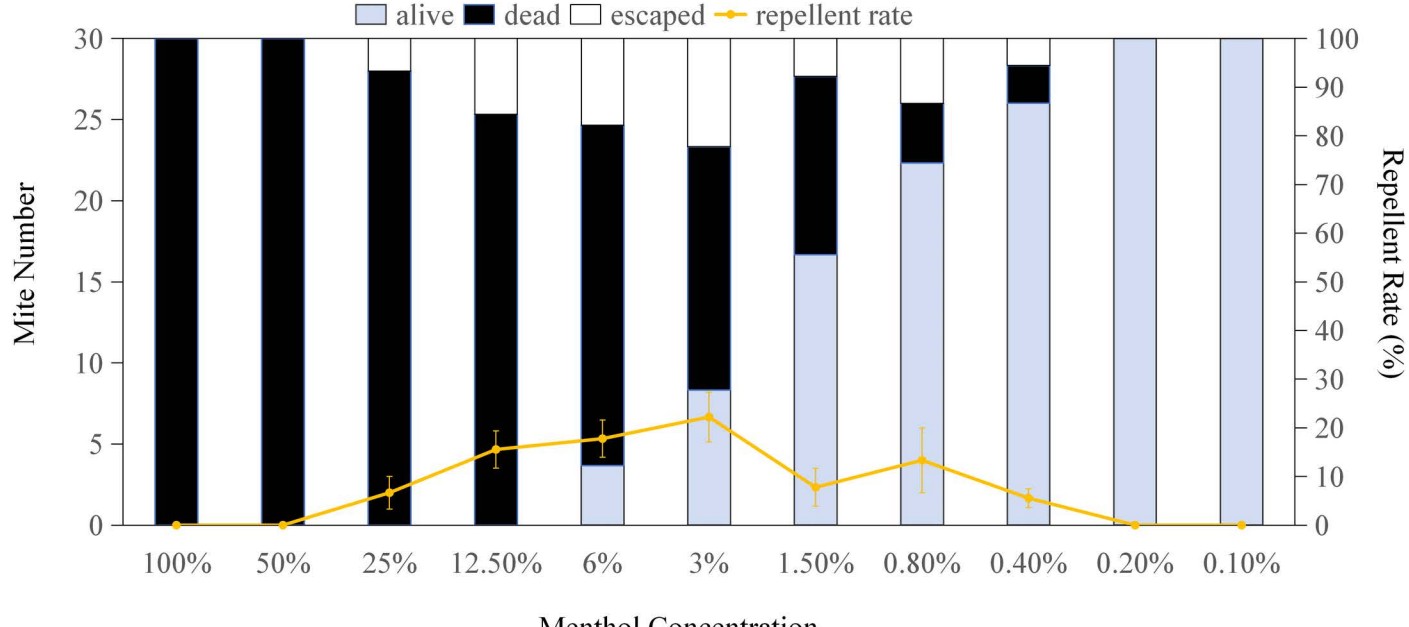

**Fig 3. Repellent effect of menthol on *Dermatophagoides pteronyssinus*.** *Dermatophagoides pteronyssinus* were exposed to menthol (100%, 50%, 25%, 12.5%, 6%, 3%, 1.5%, 0.8%, 0.4%, 0.2% or 0.1%, separately) for 2 hours. The error bars represent standard deviations.

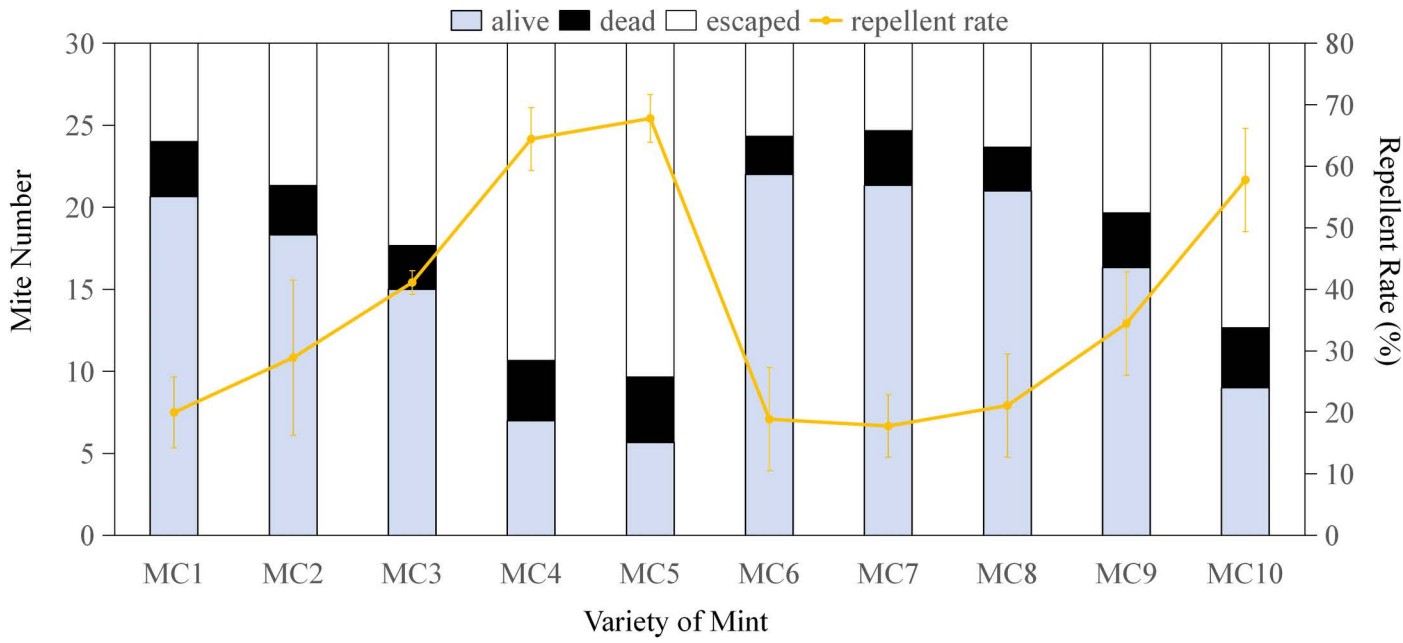

**Fig 4. Repellent effect of mint essential oils on *Dermatophagoides pteronyssinus*.** *Dermatophagoides pteronyssinus* were exposed to essential oils of 10 varieties mint (MC1-MC10) for 2 hours. The error bars represent standard deviations. MC1-MC10 represented *M. canadensis* L., *M. aquatica* Citr., *M. suaveolens* Ehrhart, *M. piperita* 'Chocolate', *M. arvensis* 'Banana', *M. japonica* (Miq.) Makino, *M. crispata* Schrader ex Willd., *M. aquatica* L., *M. piperita* 'Grapefruit' and *M. piperita* 'Candy Mint'.

collectively contribute to both behavioral (e.g., repellence, feeding deterrence) and physiological effects (e.g., acute toxicity, developmental disruption) against various pest complexes [38–42]. These acaricidal products derived from essential oils can target not only house dust mites but also plant pest mites, akin to conventional acaricides. Previous studies have extensively documented naturally occurring acaricidal constituents effective against house dust mites [43,44].

The findings of this study shed light on the potential of mint essential oils as effective agents for controlling house dust mites. The dual action of acaricidal and repellent properties exhibited by mint essential oils, along with their eco-friendly nature, positions them as promising candidates for integrated pest management strategies. Further research and field studies can explore the practical applications of mint essential oils in real-world settings to validate their efficacy as natural alternatives for house dust mite control.

## Author contributions

**Data curation:** Haiming Cai, Xu Zhang, Zhibin Lin.

**Formal analysis:** Haiming Cai, Huiquan Lin.

**Funding acquisition:** Shanshan Li, Yongwen Lin.

**Methodology:** Haiming Cai, Xu Zhang, Yongwen Lin.

**Software:** Zhibin Lin.

**Supervision:** Huiquan Lin.

**Validation:** Haiming Cai, Shanshan Li.

**Writing – original draft:** Haiming Cai, Yongwen Lin.

**Writing – review & editing:** Huiquan Lin.

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
