## [Decision Letter · Decision Letter 0]

10 Jul 2024

PONE-D-24-20445Mint Essential Oil: A Natural and Effective Agent for Controlling House Dust MitesPLOS ONE

Dear Dr. Lin,

Thank you for submitting your manuscript to PLOS ONE. After careful consideration, we feel that it has merit but does not fully meet PLOS ONE’s publication criteria as it currently stands. Therefore, we invite you to submit a revised version of the manuscript that addresses the points raised during the review process.

Dear As you can see, one of reviewers rejected your manuscript. So be accurate and give necessary details for each of comments addressed by both reviewers in order to avoid rejection during the review's 2nd round. Good luck==============================

We look forward to receiving your revised manuscript.

Kind regards,

Rachid Bouharroud

Academic Editor

PLOS ONE

Journal Requirements:

2. Thank you for stating the following financial disclosure: "This study was supported by the Doctoral fund of Zhangzhou Institute of Technology (grant number ZZYB2207), and the Special Project of Public Welfare Scientific Research Institutes of Fujian Provincial Science and Technology Department (2023R1028001)" 

Reviewers' comments:

Reviewer's Responses to Questions

**Comments to the Author**

1. Is the manuscript technically sound, and do the data support the conclusions?

Reviewer #1: Partly

Reviewer #2: No

2. Has the statistical analysis been performed appropriately and rigorously? 

Reviewer #1: Yes

Reviewer #2: No

3. Have the authors made all data underlying the findings in their manuscript fully available?

Reviewer #1: Yes

Reviewer #2: Yes

4. Is the manuscript presented in an intelligible fashion and written in standard English?

Reviewer #1: No

Reviewer #2: Yes

5. Review Comments to the Author

Reviewer #1: Comments:

General:

This is an interesting study investigating the acaricidal and repellent properties of different varieties of mint essential oils on house dust mites. While there is a substantial body of literature on the use of such essential oils and their constituents, the novelty of this study is related their potential application in managing dust mites, and identification of potential active ingredients. In general the methods appear sound and the findings worthwhile, however there are several areas where the manuscript should be improved before being considered for publication in PLOS One, which I’ve outlined below. A thorough review for english editing is also suggested.

Specific comments:

Introduction:

L47-48 – needs citation.

L50 – capitalization not required for yellow mealworm larvae

L51-52- needs citation.

L52-54 – needs citation.

L69-73 – should this section be in past, or future tense – may depend on journal format?

Methods:

L76 – When starting a sentence, genus names should always be written in full and not abbreviated. Change elsewhere, including in figure titles.

L87 – Table 1 – Abbreviations for the mint varieties are useful for later context, however the images of the plants provide nothing informative to this study. Please remove them.

L87 – Table 1 title should read ‘Ten varieties of mint used in this study and their respective abbreviations’.

L91 – change ‘A 1 kg mint sample’ to ‘Each one kg mint sample’.

L94 – list the supplier information for the sodium sulfate.

L99 – provide address of manufacturer.

L115 – Provide version and years for the spectral libraries used. What minimum confidence % was used to confirm identifications from NIST if there were no standards or Kovats indices used?

L116 – title ‘Bioassay’ is vague. Change to ‘Acaricidal bioassay’

L123 – describe how was the sex and maturity of these mites determined.

L125 – provide supplier information for benzyl benzoate.

L133 – remove ‘the’

L137 – ‘approximately three times the contact+fumigant LC100 values for each compound’ – a table listing these would be useful (or possibly add this information to Table 3 which has LC50 values?

L140 – how large was the hole in the lid? This is important as it impacts how much opportunity there will be for losing volatiles.

L141 – How were these containers maintained during the experiment? Were they in a closed container? A fume hood? What were the airflow parameters, temperature and humidity? – all factors which will influence the rate of volatile.

L148 – LC15 concentration should also be presented in a table for comparison between varieties.

L151 – “After placing the growth medium on filter paper for 5 minutes, 30 adult mites (without food) remaining on the paper were transferred to the compound-treated section using a brush to avoid damage” – I find this confusing to visualize. Suggest adding an image of the set-up showing the position of treated cloth and how the mites are added.

L153-155 – were the petri dishes covered or sealed in any way – not indicated.

L155- The term ‘escape’ suggested that a subject has exited the arena, but I believe its being used here to indicate repellency. I suggest replacing with ‘repellency’ throughout, unless the author truly means the mites are leaving the petri dish.

L159 – Add a citation for Abbott’s formula.

L167 – What statistical package was used? What was the alpha value used?

Results and Discussion:

L174 – Table 2. Title needs to be more descriptive and specific to this study. For example, instead of “Table 2. Main compound of 10 kinds of essential oils”, I suggest “Table 2: Most abundant compound identified from each of ten varieties of mint following steam distillation and GC-MS analyses”

L174 – Was there a blank solvent run of chloroform used to subtract the solvent front? Trying to determine how the % area calculations compensated for this.

L174 - Compounds need not be capitalized.

Tables 3 and 4 – alpha value for statistics should be reported.

L175 – change ‘time’ to ‘times’

L176 – Write ‘var.’ in full for a header. Also change in other locations if used in titles for tables and figures.

L192 – Table 3. I’m very concerned about the concentrations and volumes being reported for the LC50s and fumigant toxicities. It would be very useful to see the % mortality reported alongside these values (its challenging to consider any of the volumes being reported here being of realistic application – even 65ul/cm2 of MC10 would seem a massive amount of solvent for a small area).

L201 – 200ml/cm2 of MC5 oil? Is this correct? Seems physically impossible.

L219 – ‘Repellent effect of branched fatty acids’? This has nothing to with this paper?

L222-223 – “The absence of dead mites following ethanol treatment indicates that the repellent effect is attributed to the mint oils and main compounds”. Its unclear what this statement is trying to infer, but given that this is reporting a repellent assay, one would be typically using sublethal dosages. I’m also wondering if this bioassay should be names something different – since it actually investigates a range from 100% mortality to 0% repellency.

L224 – see above comment on ‘escaping’

L230 – Suggest changing the mean number of mites repelled (20.33, 19.33, 17.33) to a percentage instead.

L234 – spelling “subletial’ = ‘sublethal’

L235 – “The strong aroma of mint may disrupt the sensory receptors

of house dust mites, deterring their presence and reducing infestation risks” – this is a very vague statement, and needs further context and explanation.

References: - needs a thorough check for formatting consistency, as well as correct formatting for latin names

Figures:

- Indicate the LD50 on the figures.

Other points:

- The presentation and interpretation of the repellent bioassay is insufficient. More comparison and detail is warranted.

- The discussion should be expanded to include points on:

o Potential mode of action of the essential oil constituents (active ingredients) which were investigated. There are many papers which investigate the effects of these compounds (i.e. linalool, menthol, limonene) on other species. Discussing the known lethal and sublethal effects on other arthropods, and acarines, would be important for context.

o More detail on proposed application of this knowledge, or key next steps should be included.

Reviewer #2: I read the manuscript (PONE-D-24-20445) entitled "Mint Essential Oil: A Natural and Effective Agent for Controlling House Dust Mites," written by Cai et al. for publication in PLOS ONE. The study explores the effectiveness of mint essential oils in controlling house dust mites. The research title is interesting and might attract the reader's interest; however, the methodology and results sections are poorly written. In particular, the author should consider statistical analysis to provide robust evidence for their results. The manuscript has no result interpretation, comparative discussion about their results, or implications. I think this manuscript is not suitable for publication in its current form. Please see some minor comments and suggestions below.

Lines 46–49: Please provide references for these statements.

Lines 51–54: Please provide references for these statements.

In table 1, what is “M. × piperita”?

Extraction of Essential Oils; Please briefly explain the methods and materials used for essential oil extraction.

Why did the author use 5-8-day-old adults for bioassay? What is the average adult longevity for this species? What is the ratio of males and females used in the bioassay?

What concentration of mint oil was used for the experiment? How did the author prepare the stock solutions? The author needs to provide this information through materials and methods.

Why was benzyl benzoate used as a positive control? What concentration was used for the bioassay? Please provide information about benzyl benzoate in the chemicals section.

Please utilize the statistical analysis in Table 3 to see the significant differences in LC50 values among the varieties.

Were there any significant differences in mortality between the varieties?

6. PLOS authors have the option to publish the peer review history of their article (what does this mean? ). If published, this will include your full peer review and any attached files.

**Do you want your identity to be public for this peer review?** For information about this choice, including consent withdrawal, please see our Privacy Policy .

Reviewer #1: No

Reviewer #2: No

---

## [Author Response · Author response to Decision Letter 1]

9 Aug 2024

Point-by-point responses to Reviewers and Editor

Reviewer #1: Comments:

Introduction:

L47-48 – needs citation.

Respond: The citation is included as suggested.

L50 – capitalization not required for yellow mealworm larvae

Respond: Corrected.

L51-52- needs citation.

Respond: Citation included as suggested.

L52-54 – needs citation.

Respond: Done as suggested.

L69-73 – should this section be in past, or future tense – may depend on journal format?

Respond: The section has been revised accordingly.

Methods:

L76 – When starting a sentence, genus names should always be written in full and not abbreviated. Change elsewhere, including in figure titles.

Respond: Many thanks for this observation. This has been corrected throughout the manuscript as suggested.

L87 – Table 1 – Abbreviations for the mint varieties are useful for later context, however the images of the plants provide nothing informative to this study. Please remove them.

Respond: Revised as suggested.

L87 – Table 1 title should read ‘Ten varieties of mint used in this study and their respective abbreviations’.

Respond: Corrected as suggested.

L91 – change ‘A 1 kg mint sample’ to ‘Each one kg mint sample’.

Respond: Done as suggested.

L94 – list the supplier information for the sodium sulfate.

Respond: Supplier information provided as suggested.

L99 – provide address of manufacturer.

Respond: Provided as suggested.

L115 – Provide version and years for the spectral libraries used. What minimum confidence % was used to confirm identifications from NIST if there were no standards or Kovats indices used?

Respond: NIST 2017. It was added in the revised version of the manuscript to capture that as suggested.

L116 – title ‘Bioassay’ is vague. Change to ‘Acaricidal bioassay’

Respond: Done as suggested.

L123 – describe how was the sex and maturity of these mites determined.

Respond: We did not distinguish the sex of mites, and the maturity was determined by the exuviate time.

L125 – provide supplier information for benzyl benzoate.

Respond: Provided as suggested.

L133 – remove ‘the’

Respond: Deleted as suggested.

L137 – ‘approximately three times the contact+fumigant LC100 values for each compound’ – a table listing these would be useful (or possibly add this information to Table 3 which has LC50 values?

Respond: The information is now added to Table 3 as suggested.

L140 – how large was the hole in the lid? This is important as it impacts how much opportunity there will be for losing volatiles.

Respond: 0.5 cm diameter, and this is now added in the revised version of the manuscript.

L141 – How were these containers maintained during the experiment? Were they in a closed container? A fume hood? What were the airflow parameters, temperature and humidity? – all factors which will influence the rate of volatile.

Respond: Kept in a closed container and the information is now added in the revised version of the manuscript as suggested.

L148 – LC15 concentration should also be presented in a table for comparison between varieties.

Respond: Addressed as suggested, and this is now included in Table 3.

L151 – “After placing the growth medium on filter paper for 5 minutes, 30 adult mites (without food) remaining on the paper were transferred to the compound-treated section using a brush to avoid damage” – I find this confusing to visualize. Suggest adding an image of the set-up showing the position of treated cloth and how the mites are added.

Respond: We revised the statement for more clarity and easy to visualize.

L153-155 – were the petri dishes covered or sealed in any way – not indicated.

Respond: Yes, they were covered; this is now captured in the revised version of the manuscript to avoid confusion.

L155- The term ‘escape’ suggested that a subject has exited the arena, but I believe its being used here to indicate repellency. I suggest replacing with ‘repellency’ throughout, unless the author truly means the mites are leaving the petri dish.

Respond: Many thanks for this remark. Changed to repellency throughout the text as suggested.

L159 – Add a citation for Abbott’s formula.

Respond: Done as suggested.

L167 – What statistical package was used? What was the alpha value used?

Respond: Graphpad Prism 9.0 software were used and now captured in the revised version of the manuscript.

Results and Discussion:

L174 – Table 2. Title needs to be more descriptive and specific to this study. For example, instead of “Table 2. Main compound of 10 kinds of essential oils”, I suggest “Table 2: Most abundant compound identified from each of ten varieties of mint following steam distillation and GC-MS analyses”

Respond: Done as suggested.

L174 – Was there a blank solvent run of chloroform used to subtract the solvent front? Trying to determine how the % area calculations compensated for this.

Respond: No, we did not use blank solvent.

L174 - Compounds need not be capitalized.

Tables 3 and 4 – alpha value for statistics should be reported.

Respond: Corrected as suggested.

L175 – change ‘time’ to ‘times’

Respond: Changed as suggested.

L176 – Write ‘var.’ in full for a header. Also change in other locations if used in titles for tables and figures.

Respond: Done as suggested.

L192 – Table 3. I’m very concerned about the concentrations and volumes being reported for the LC50s and fumigant toxicities. It would be very useful to see the % mortality reported alongside these values (its challenging to consider any of the volumes being reported here being of realistic application – even 65ul/cm2 of MC10 would seem a massive amount of solvent for a small area).

Respond: Revised as suggested.

L201 – 200ml/cm2 of MC5 oil? Is this correct? Seems physically impossible.

Respond: Sorry for this mistake. That is 200 μl/cm2 and it is now corrected in the revised version of the manuscript.

L219 – ‘Repellent effect of branched fatty acids’? This has nothing to with this paper?

Respond: Sorry for this mistake from our end. This should rather read ‘Repellent effect of main compounds of mint’. It is now corrected in the revised manuscript.

L222-223 – “The absence of dead mites following ethanol treatment indicates that the repellent effect is attributed to the mint oils and main compounds”. Its unclear what this statement is trying to infer, but given that this is reporting a repellent assay, one would be typically using sublethal dosages. I’m also wondering if this bioassay should be names something different – since it actually investigates a range from 100% mortality to 0% repellency.

Respond: The section is revised accordingly for more clarity and avoid confusion.

L224 – see above comment on ‘escaping’

Respond: Addressed accordingly.

L230 – Suggest changing the mean number of mites repelled (20.33, 19.33, 17.33) to a percentage instead.

Respond: Corrected as suggested.

L234 – spelling “subletial’ = ‘sublethal’

Respond: Corrected.

L235 – “The strong aroma of mint may disrupt the sensory receptors

of house dust mites, deterring their presence and reducing infestation risks” – this is a very vague statement, and needs further context and explanation.

Respond: The sentence is now revised for more clarity.

References: - needs a thorough check for formatting consistency, as well as correct formatting for latin names

Respond: The references have been revised as per the guidelines of the journal for uniformity.

Figures:

- Indicate the LD50 on the figures.

Respond: Done.

Other points:

- The presentation and interpretation of the repellent bioassay is insufficient. More comparison and detail is warranted.

Respond: Done!

- The discussion should be expanded to include points on:

o Potential mode of action of the essential oil constituents (active ingredients) which were investigated. There are many papers which investigate the effects of these compounds (i.e. linalool, menthol, limonene) on other species. Discussing the known lethal and sublethal effects on other arthropods, and acarines, would be important for context.

Respond: Done!

o More detail on proposed application of this knowledge, or key next steps should be included.

Respond: Done!

Reviewer #2:

Lines 46–49: Please provide references for these statements.

Respond: Reference included as suggested.

Lines 51–54: Please provide references for these statements.

Respond: Done as suggested.

In table 1, what is “M. × piperita”?

Respond: That is Mentha piperita. It is now corrected in the revised version of the manuscript.

Extraction of Essential Oils; Please briefly explain the methods and materials used for essential oil extraction.

Respond: More explanation has been added as suggested (lines xxxxx).

Why did the author use 5-8-day-old adults for bioassay? What is the average adult longevity for this species? What is the ratio of males and females used in the bioassay?

Respond: According to Kim JR, Perumalsamy H, Kwon MJ, Chae SU, Ahn YJ. Toxicity of hiba oil constituents and spray formulations to American house dust mites and copra mites. Pest Manag Sci. 2015 May;71(5):737-43, the average of adult longevity of house dust mite is about 20 days! That is why we selected one week old adult for the bioassay for easy manipulation and avoid physical damage. We did not distinguish the sex for mite during the bioassays.

What concentration of mint oil was used for the experiment? How did the author prepare the stock solutions? The author needs to provide this information through materials and methods.

Respond: Various concentrations of each test substance in 50 μL of ethanol were administered to 4.5 cm diameter black cotton-fabric circles, resulting in mint oil quantities of 300, 150, 75, 37.5, 18.75, 9.325, and 5 μL/cm² applied on the fabric. So, these were the various concentrations used, and they are captured in the materials and methods section of the revised manuscript. We used the pure mint oil directly which was consider as the stock solution too.

Why was benzyl benzoate used as a positive control? What concentration was used for the bioassay? Please provide information about benzyl benzoate in the chemicals section.

Respond: Benzyl benzoate was an effective recommended acaricide, and it was used in many previous studies. This is now included in the revised version of the manuscript.

Please utilize the statistical analysis in Table 3 to see the significant differences in LC50 values among the varieties.

Respond: Many thanks for the observation. However, LC50 is an estimated data rather than a measured data, and it is an estimated value of several sets of repeated test data, which is generally generated and not used for statistical difference analysis.

Were there any significant differences in mortality between the varieties?

Respond: Not yet observed as per our results. However, we can design further experiments as per your comment to investigate further on this aspect.

---

## [Editor Report · Decision Letter 1]

28 Aug 2024

PONE-D-24-20445R1Mint Essential Oil: A Natural and Effective Agent for Controlling House Dust MitesPLOS ONE

Dear Dr. Lin,

Thank you for submitting your manuscript to PLOS ONE. After careful consideration, we feel that it has merit but does not fully meet PLOS ONE’s publication criteria as it currently stands. Therefore, we invite you to submit a revised version of the manuscript that addresses the points raised during the review process.

We look forward to receiving your revised manuscript.

Kind regards,

Rachid Bouharroud

Academic Editor

PLOS ONE

Journal Requirements:

Additional Editor Comments:**Dear Authors****I read carefully your revised manuscript and now looks better than the 1st version.****However still one comment related to LC50 significance that I'm not agree with your feedback. LC/LD50 is "estimated" but obtained from measured data and each measured data should be statistically checked. Refer to many papers dealing with LD50 significance.****Another comment related to editing. Please let me know the lines refered to this feedback (** Extraction of Essential Oils; Please briefly explain the methods and materials used for essential oil extraction.

Respond: More explanation has been added as suggested (lines xxxxx).** )****Good luck**

---

## [Author Response · Author response to Decision Letter 2]

10 Jan 2025

Point-by-point responses to Reviewers and Editor

Comment: Journal Requirements:

Responding: Thank you for you comment! We had check all the references and did not change this part.

Additional Editor Comments:

Comment 1: However still one comment related to LC50 significance that I'm not agree with your feedback. LC/LD50 is "estimated" but obtained from measured data and each measured data should be statistically checked. Refer to many papers dealing with LD50 significance.

Responding: Thank you for your kindly! However, we did not want to show or describe the significant different of LC50, we just mentioned the level.

Comment 2: Another comment related to editing. Please let me know the lines refered to this feedback (Extraction of Essential Oils; Please briefly explain the methods and materials used for essential oil extraction.)

Responding: I am sorry for puzzling you. Please check Line 92-95 in the clean version.

---

## [Editor Report · Decision Letter 2]

21 Jan 2025

Mint Essential Oil: A Natural and Effective Agent for Controlling House Dust Mites

PONE-D-24-20445R2

Dear Dr. Lin,

We’re pleased to inform you that your manuscript has been judged scientifically suitable for publication and will be formally accepted for publication once it meets all outstanding technical requirements.

Kind regards,

Rachid Bouharroud

Academic Editor

PLOS ONE
---

## [Editor Report · Acceptance letter]

PONE-D-24-20445R2

PLOS ONE

Dear Dr. Lin,

I'm pleased to inform you that your manuscript has been deemed suitable for publication in PLOS ONE. Congratulations! Your manuscript is now being handed over to our production team.

Kind regards,

on behalf of

Dr. Rachid Bouharroud

Academic Editor

PLOS ONE
